# A recurrent regulatory change underlying altered expression and Wnt response of the stickleback armor plates gene *EDA*

Natasha M O'Brown[1], Brian R Summers[1†], Felicity C Jones[1‡], Shannon D Brady[1,2], David M Kingsley[1,2*]

[1]Department of Developmental Biology, Stanford University School of Medicine, Stanford, United States; [2]Howard Hughes Medical Institute, Stanford University School of Medicine, Stanford, United States

**Abstract** Armor plate changes in sticklebacks are a classic example of repeated adaptive evolution. Previous studies identified *ectodysplasin (EDA)* gene as the major locus controlling recurrent plate loss in freshwater fish, though the causative DNA alterations were not known. Here we show that freshwater *EDA* alleles have *cis*-acting regulatory changes that reduce expression in developing plates and spines. An identical T → G base pair change is found in *EDA* enhancers of divergent low-plated fish. Recreation of the T → G change in a marine enhancer strongly reduces expression in posterior armor plates. Bead implantation and cell culture experiments show that Wnt signaling strongly activates the marine *EDA* enhancer, and the freshwater T → G change reduces Wnt responsiveness. Thus parallel evolution of low-plated sticklebacks has occurred through a shared DNA regulatory change, which reduces the sensitivity of an *EDA* enhancer to Wnt signaling, and alters expression in developing armor plates while preserving expression in other tissues.

*For correspondence: kingsley@stanford.edu

Present address: †General Practice Dentistry, Albany, United States; ‡Friedrich Miescher Laboratory, Max Planck Institute for Developmental Biology, Tuebingen, Germany

Competing interests: The authors declare that no competing interests exist.

## Introduction

The repeated evolution of similar adaptive phenotypic traits in multiple populations is a fascinating evolutionary phenomenon observed in many organisms (*Wood et al., 2005*; *Elmer and Meyer, 2011*; *Conte et al., 2012*; *Martin and Orgogozo, 2013*; *Stern, 2013*). The threespine stickleback (*Gasterosteus aculeatus*) is a particularly favorable species to characterize the molecular mechanisms underlying repeated evolution of adaptive phenotypic traits in nature, because many populations have evolved similar morphological and skeletal traits following widespread colonization of new freshwater environments by migratory marine ancestors at the end of the last ice age (*Bell and Foster, 1994*).

One of the most striking and ubiquitous morphological changes seen in sticklebacks is repeated alteration in bony armor seen along the sides of fish. Marine sticklebacks are typically covered from head to tail with 32 (or more) bony lateral plates. In contrast, freshwater fish characteristically lack most plates, typically retaining only 0–7 plates in the anterior flank region. This dramatic difference in anterior-posterior patterning of armor plates was used to assign different species names to marine and freshwater sticklebacks in the 1800s (*Cuvier and Valenciennes, 1829*). Subsequent studies have shown that different armor patterns are highly heritable, and are likely controlled by a relatively simple genetic system (*Münzing, 1959*; *Hagen and Gilbertson, 1973*; *Avise, 1976*; *Ziuganov, 1983*; *Banbura, 1994*). More recently, genome-wide linkage mapping in crosses between divergent sticklebacks identified a major locus on stickleback chromosome IV that controls over 75% of the variance in armor plate number in F2 offspring (*Colosimo et al., 2004*; *Cresko et al., 2004*), as well as several unlinked modifier genes that each control 5–10% of the variance in plate numbers (*Colosimo et al., 2004*).

**eLife digest** Stickleback fish develop bony plates on their surface to protect themselves from predators. The extent and pattern of their bony armor depends on their habitat: marine sticklebacks are typically covered from head to tail with bony plates, but freshwater sticklebacks retain only a few plates on their sides.

One gene that promotes the formation of the bony plates is called *ectodysplasin* (*EDA*). This encodes a signaling protein that is important for the development of the skeleton, skin and many other tissues. Variations in the sequence of this gene are shared among different stickleback populations worldwide. However, it has not been clear which genetic changes can explain how lightly armored freshwater sticklebacks could have evolved from their well-armored marine ancestors on several separate occasions.

Here, O'Brown et al. studied *EDA* in marine and groups of freshwater sticklebacks that have evolved in different locations around the world. The experiments show that the level of expression of *EDA* in the developing plates and spines is lower in the freshwater fish. O'Brown et al. thought this could be due to genetic changes in regions of *EDA* that lie outside the region that encodes the protein, so called 'regulatory elements'.

Indeed, further experiments found that all freshwater fish have a small change in the DNA of a regulatory element that switches on the gene in plate-forming regions of the body. When this change was introduced into marine sticklebacks, the fish had lower levels of *gene* expression in these plate-forming regions.

These findings demonstrate that lightly armored sticklebacks have evolved multiple times from their well-armored marine ancestors through the same small change in their DNA that alters the expression of the *EDA* gene. The next challenge will be to understand why this particular small change in DNA appears to be favored over all the other changes that could occur in the regulatory element, and to see if factors that act through this regulatory switch also modify armor structures in natural populations.

High-resolution mapping, chromosome walking, and transgenic rescue experiments showed that the major armor plate locus corresponds to the *ectodysplasin* (*EDA*) gene on stickleback chromosome IV (*Colosimo et al., 2005*). The *EDA* gene encodes a secreted protein in the tumor necrosis factor (TNF) family that plays a key role in cell signaling during the development of multiple neural crest and ectodermal tissues, including skin, hair, and teeth (*Mikkola and Thesleff, 2003*; *Cui and Schlessinger, 2006*). Humans with null mutations in *EDA* have defects in multiple ectoderm and neural crest derived tissues, including sparse hair, absent sweat glands, dental abnormalities, and dermal bone changes in the skull (*Mikkola and Thesleff, 2003*; *Cui and Schlessinger, 2006*; *Yavuz et al., 2006*; *Clauss et al., 2008*; *Lesot et al., 2009*). Zebrafish and medaka mutants with perturbations in the EDA pathway display severe skeletal abnormalities, such as loss of fins and scales, missing and abnormally shaped teeth, and abnormal craniofacial morphology (*Harris et al., 2008*; *Iida et al., 2014*).

While both high-resolution mapping and transgenic rescue experiments confirm that *EDA* is the major locus controlling armor plates in sticklebacks, the molecular difference between marine and freshwater fish is still unclear. Most freshwater populations share four amino acid differences in the *EDA* gene, as well as numerous non-coding changes that together make up a characteristic freshwater haplotype (*Colosimo et al., 2005*). However, the four amino acid changes occur at positions that also vary among other species, so these coding changes are unlikely to be the basis of major changes in EDA function. In addition, there exists at least one low-plated stickleback population that has the identical EDA protein-coding sequence as marine fish (*Colosimo et al., 2005*). This key population from Nakagawa Creek in Gifu, Japan (NAKA) is a low-plated stream population with a predominately marine-like sequence in both coding and non-coding regions. NAKA fails to complement armor plate changes when crossed with a typical Canadian low-plated population (*Schluter et al., 2004*), suggesting that NAKA and other low-plated fish share a modification in the same major locus. Based on the absence of amino acid changes in NAKA, and the deleterious nature of coding region changes in human patients, *Colosimo et al. (2005)* proposed that an unknown regulatory change at the stickleback *EDA* locus is the most likely basis of the common *EDA* variants found in freshwater fish.

Here we further investigate the *EDA* locus in order to study the causative base pair changes that underlie repeated evolution of low-plated sticklebacks.

## Results

### A *cis*-acting regulatory change reduces expression of freshwater *EDA* gene

In order to test if *EDA* is differentially expressed in marine and freshwater fish due to *cis*-regulatory differences, we performed allele-specific expression in F1 hybrid fish made by crossing marine and freshwater sticklebacks. The F1 hybrids are heterozygous for both the marine and freshwater haplotypes at the *EDA* locus, and therefore express both alleles in an identical *trans*-acting environment. We then isolated RNA from 10 different developing tissues, and determined whether the freshwater and marine *EDA* transcripts were expressed at the same or different levels using pyrosequencing (*Figure 1*, see 'Materials and methods'). No significant expression differences between marine and freshwater *EDA* alleles were observed in the fins or the lower jaw. However, the freshwater *EDA* allele was expressed

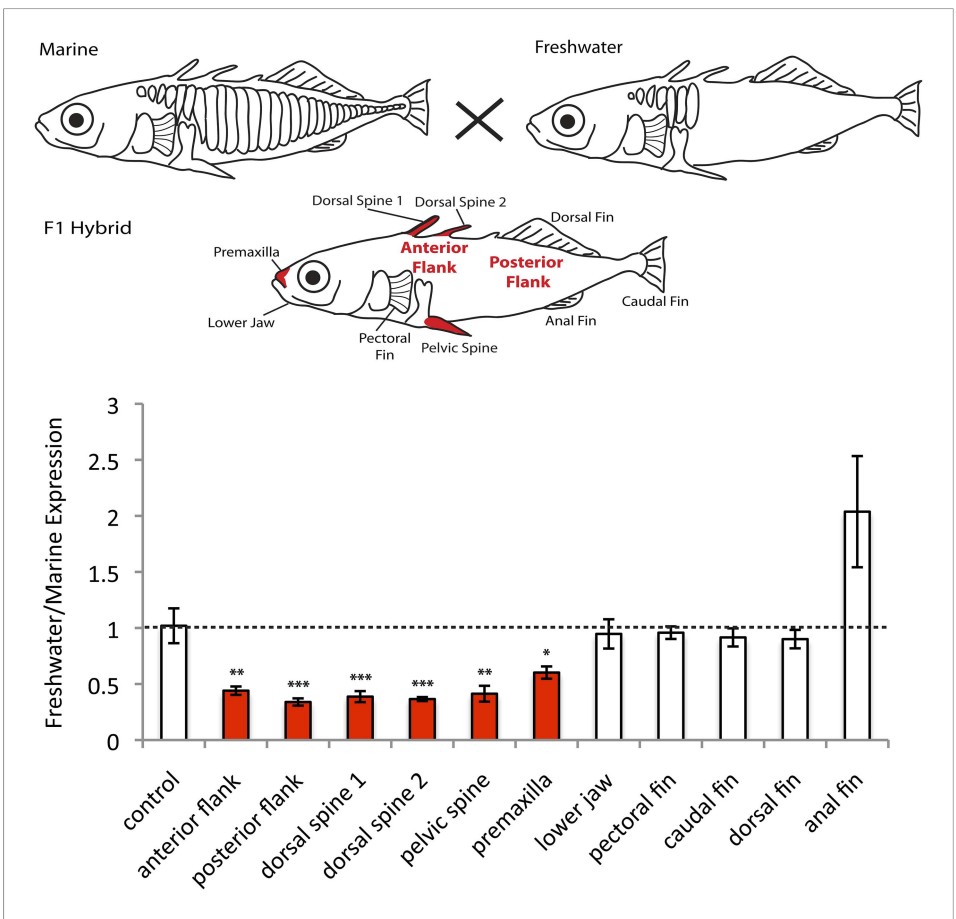

**Figure 1**. *EDA* shows allele-specific expression differences in several tissues, indicating *cis*-regulatory divergence. Allele-specific expression in F1 freshwater-marine heterozygous larvae reveals significant differential expression of the marine and freshwater alleles in dorsal spines 1 and 2, the pelvic spine, the premaxilla, and the presumptive armor plates (anterior and posterior flanks). In all of these bony tissues the marine allele of *EDA* is expressed more highly than the freshwater allele, suggesting that there are differences in the *cis*-regulatory sequences controlling *EDA* expression. Several other tissues, however, do not show significant allelic imbalance in *EDA* expression; their allelic ratios are close to 1 (dashed line). The control shows results from a 1:1 mixture of plasmids containing the freshwater and marine alleles. Red-shaded structures and bars indicate tissues with significant allelic-imbalance compared to control (***p < 0.001, **p < 0.01, *p < 0.05 by two-tailed t-test).

almost fourfold lower than the marine allele in the developing anterior and posterior flanks (corresponding to sites where armor plates had already appeared, or were not forming yet; respectively), and in the dorsal and pelvic spines (p < 0.01, Student's *t* test), as well as twofold lower in the premaxilla (p < 0.05, Student's *t* test). These data suggest that the marine and freshwater haplotypes at the *EDA* locus have *cis*-acting regulatory changes that reduce expression of the freshwater allele in particular tissues, including the flank regions where armor plates normally form.

## Identification of a single base pair change shared by all low-plated fish

Previous studies narrowed the minimal candidate interval controlling armor plates to a 16 kb interval containing *EDA* and flanking regions (*Colosimo et al., 2005*). To look for possible shared molecular changes that might account for the regulatory difference between marine and freshwater sticklebacks, we amplified and sequenced the *EDA* candidate interval from low-plated Japanese NAKA fish, and compared it to other high- and low-plated stickleback populations (*Figure 2* and 'Materials and methods'). This analysis identified a single T → G nucleotide change, located at position chrIV: 12811481 (gasAcu1 assembly, *Jones et al., 2012*) in the intergenic region downstream of *EDA*, that was shared between NAKA and all other low-plated sticklebacks examined.

## The low-plated base pair change alters the activity of an *EDA* enhancer

Given the known role of *EDA* in plate formation, we hypothesized that this intergenic base pair change (*Figure 2*) lies in a developmental enhancer that modulates *EDA* gene expression during armor plate development. Therefore, we cloned a 3.2 kb region surrounding the SNP (orange bar, *Figure 2*) from high-plated marine fish and tested for enhancer activity using a GFP reporter construct

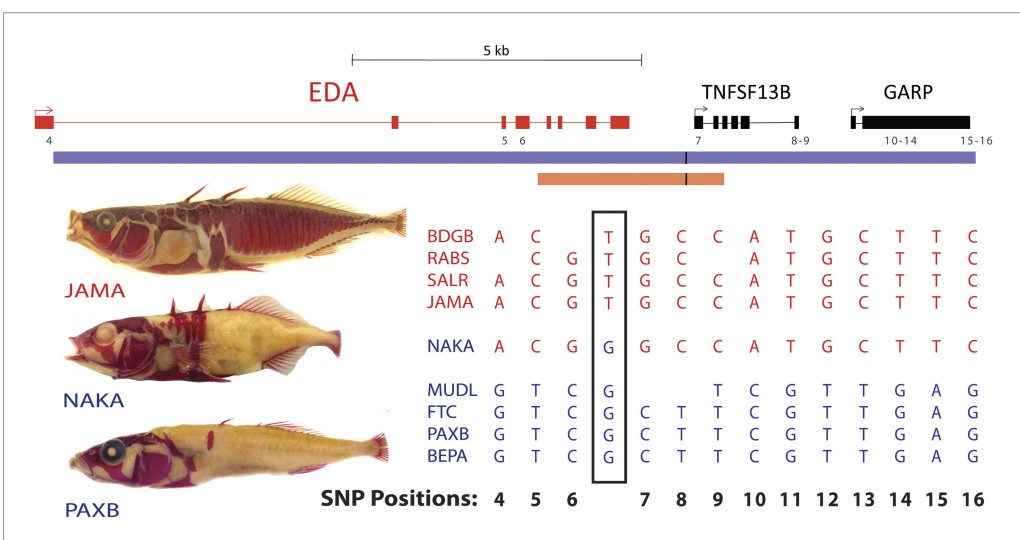

**Figure 2**. All low-plated populations share a single base pair change in the genetic region controlling armor plates. Genome-wide comparisons of low- and high-plated fish reveal a T → G base pair change (black box) that is shared between all low-plated populations tested, including the low-plated Japanese NAKA fish that otherwise shows a primarily marine-like haplotype in the *EDA* region. Geographic population codes and DNA sequences from marine high-plated populations and freshwater low-plated populations are shown in red and blue, respectively, along with representative Alizarin Red stained fish showing typical armor plate patterns in different fish. The 16 kb candidate interval controlling armor plate number (blue bar, *Colosimo et al., 2005*) is shown beneath predicated genes in the region. Also shown are the numbered positions (4–16) of previously identified SNPs that differentiate most low- and high-plated sticklebacks other than NAKA (*Colosimo et al., 2005*). These numbered SNPs correspond to positions chrIV: 12800508, 12808303, 12808630, 12811933, 12813328, 12813394, 12815024, 12815027, 12816201, 12816202, 12816360, 12816402, and 12816464 in the stickleback genome assembly (*Jones et al., 2012*). Blank positions represent occasional gaps in sequence coverage for individual fish from large population surveys (*Colosimo et al., 2005*; *Jones et al., 2012*). The position of the shared T → G change (chrIV:12811481) is indicated with a short black vertical line in the overall genomic interval, and in a 3.2 kb region that was used to test for possible regulatory enhancers in the *EDA* region (orange bar, chrIV:12808949–12812120).

(p3.2mar-GFP, see 'Materials and methods'). In two-month-old transgenic fish, this 3.2 kb region drives consistent GFP expression at multiple sites, including the anterior and posterior armor plates, the junction between the pelvic spine and girdle, the upper edge of the pelvic girdle, the base of the pectoral fin, the cranial ganglia surrounding the eyes and lips, and the premaxilla and jaw (*Figure 3*, *Figure 4A,B*). Comparison to the endogenous pattern of *EDA* expression using in situ hybridization suggests that the GFP construct recapitulates typical *EDA* patterns in cranial ganglia, premaxilla, jaw, pectoral fin base, armor plates, and pelvic girdle base (*Figure 3*). However, some domains of endogenous *EDA* expression are not accounted for by the enhancer region, including the dorsal and pelvic spines, suggesting that this construct contains some but not all of the regulatory information controlling *EDA* expression during normal development.

We next performed site-directed mutagenesis to change the T found in high-plated fish to the G found in all sequenced low-plated fish, while maintaining the sequence of the high-plated marine haplotype throughout the rest of the enhancer construct. The p3.2mar(T → G)-GFP plasmid still drove detectable expression in the anterior plates, cranial ganglia, jaws, and pectoral fin base, but showed greatly reduced GFP expression in the posterior armor plates and pelvic girdle junction (*Figure 4C,D*, *Figure 4—figure supplement 1*). Thus, the single base pair change shared by all low-plated sticklebacks produces striking but localized differences in gene expression, with prominent reduction occurring in the flank region where plates normally develop in marine fish.

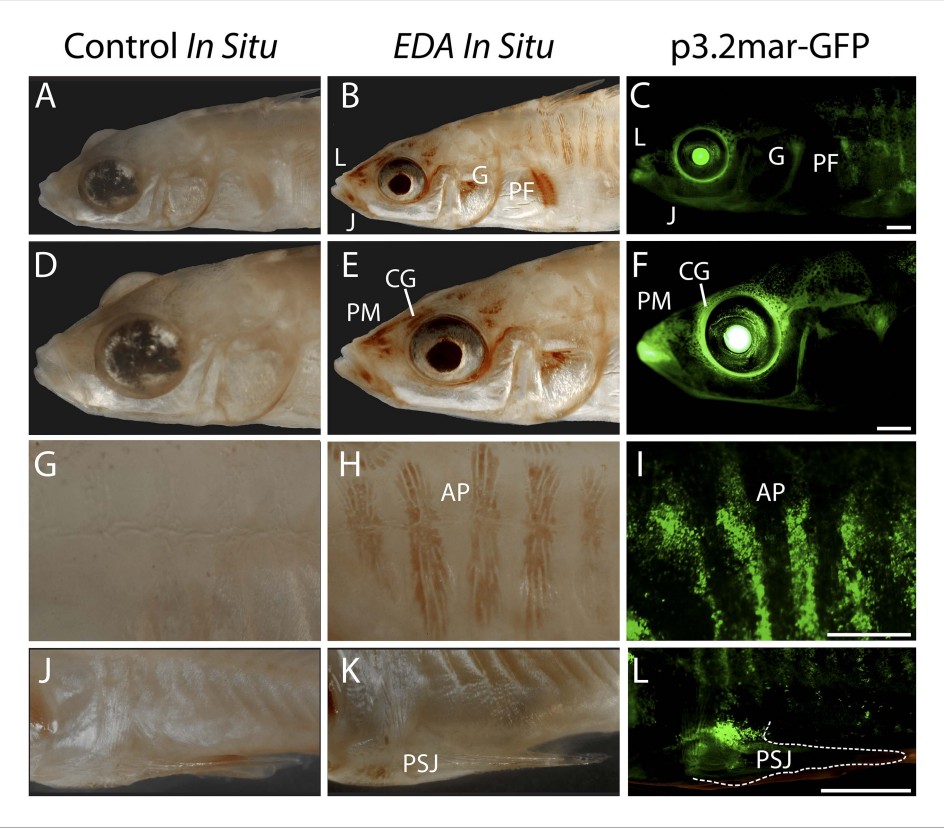

**Figure 3**. Reporter expression driven by an *EDA* enhancer matches several regions of endogenous *EDA* expression. (**A**, **D**, **G**, **J**) Negative control DapB RNAscope in situ staining shows no positive brown signal appearing around the face (**A** and **D**), the plates (**G**), or the pelvic junction (**J**). The slight brown color in the pelvic spine is due to natural pigmentation at this site. (**B**, **E**, **H**, **K**) Endogenous *EDA* expression is localized to the premaxilla, lips, lower jaw, cranial ganglia, gill and pectoral fin base (**B** and **E**); armor plates (**H**); and the junction between the pelvic spine and the pelvic girdle (**K**). (**C**, **F**, **I**, **L**) The p3.2mar-GFP construct drives reporter expression at several corresponding sites, including the lips, premaxilla, lower jaw and cranial ganglia surrounding the eyes (**C** and **F**); in the armor plates; (**I**) and at the pelvic junction (**L**). Anatomical abbreviations as in other figures, including: lips (L), premaxilla (PM), lower jaw (J), cranial ganglia (CG), gills (G), pectoral fin base (PF), anterior plates (AP), and pelvic spine junction (PSJ). Scale bars are 1 mm long.

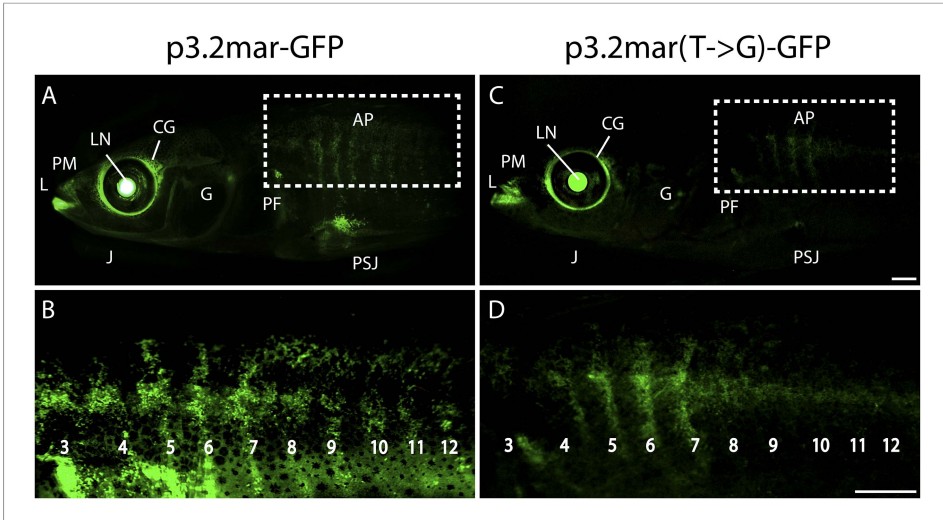

**Figure 4**. Enhancer expression in plates and other structures is reduced by a single base pair change. (**A**, **B**) A 3.2 kb enhancer region from high-plated fish drives GFP expression in all armor plates (AP) of 2-month-old (20 mm long) marine stickleback larvae, with expression preceding plate ossification, and stronger expression in the first 7 armor plates. The p3.2mar-GFP construct also drives expression in the lips (L), premaxilla (PM), lower jaw (J), cranial ganglia (CG), the base of the pectoral fins (PF), and the pelvic spine-girdle junction (PSJ). Panel **B** is a higher magnification view of the area boxed in panel **A**. (**C**, **D**) The single base pair change in the p3.2mar(T → G)-GFP construct results in greatly reduced enhancer activity in the posterior plates, and reduced but detectable expression in plates 4–7 (**D**). This stable line also retains expression in the cranial ganglia and lips, reduced expression in the pelvic junction and the pectoral fin base, and novel strong expression in the spinal cord. Panel **D** is a higher magnification view of the area boxed in panel **B**. The hsp70 promoter in the GFP vector drives strong expression in the lens (LN) of all transgenic fish, helping to identify carriers following microinjection experiments (*Chan et al., 2010*). Scale bars are 1 mm long.

The following figure supplement is available for figure 4:

**Figure supplement 1**. Plate enhancer activity is altered by a single base pair change (additional examples from independent transgenic fish).

## Altered Wnt responsiveness of the marine and freshwater *EDA* enhancer

Previous studies have shown that Wnt signaling acts upstream of *EDA* in the early proliferation and specification of tissues in many vertebrates (*Laurikkala et al., 2002*; *Cui and Schlessinger, 2006*; *Häärä et al., 2011*; *Arte et al., 2013*). To test whether Wnt also acts upstream of plate development in sticklebacks, we tested whether implants of either Wnt-3a or Dkk-1 (an inhibitor of Wnt signaling, *Glinka et al., 1998*) altered normal patterns of armor plate formation. Beads soaked in PBS, Wnt-3a, or Dkk-1 proteins were surgically implanted into the mid-flank of 2-month-old marine fish, and fish were then aged to 6 months to test for effects on plate size and number. Control bead implantation had no significant effect on overall plate morphology (*Figure 5A,B*). In contrast, exposure to ectopic Wnt signaling at the juvenile stage induced hypermorphic plate development, characterized by adult fish with larger plates and plate fusions surrounding the sites of Wnt-3a bead implantation (*Figure 5C*). Conversely, the addition of the Wnt inhibitor Dkk-1 resulted in a hypomorphic phenotype marked by the absence of plates surrounding the bead implantation site (*Figure 5D*), suggesting that Wnt signaling plays an important role in normal plate development.

To examine whether ectopic Wnt signaling also causes changes in *EDA* expression, we placed Wnt-3a protein beads into marine fish and used in situ hybridization to visualize *EDA* expression 48 hr later (*Figure 6A,B*). These experiments revealed a strong ring of induced *EDA* expression surrounding the site of Wnt-3a bead implantation (*Figure 6B*). We then implanted Wnt-3a beads into a stable transgenic line carrying the p3.2mar-GFP reporter construct described above. Implanted Wnt-3a beads, but not control beads, induced a strong ring of GFP expression directly around the site of bead

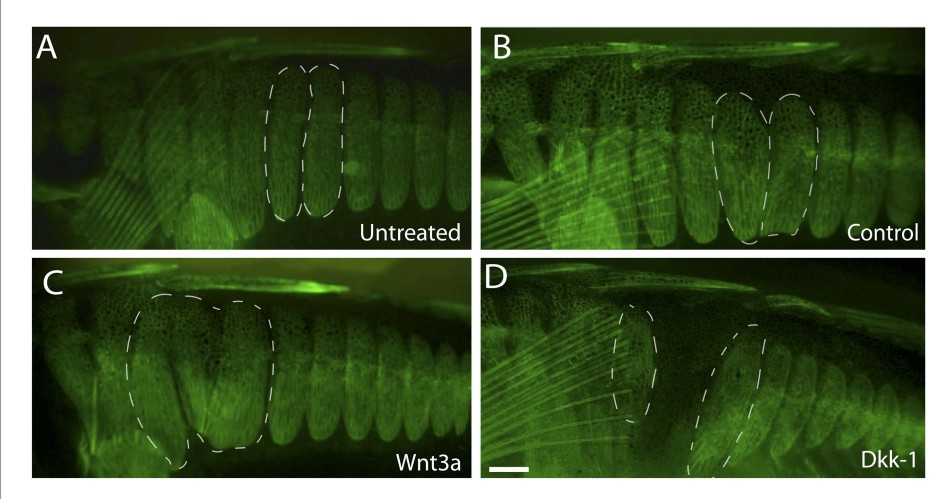

**Figure 5**. Wnt signaling regulates armor plate development. Live Calcein staining of 6-month-old fish marks newly ossified bones in green. (**A**) Armor plates in an untreated high-plated adult marine fish. The normal morphologies of two individual plates are outlined with dashed lines. (**B**) Control beads soaked in PBS were implanted between the two outlined plates at two months of age. After bead implantation, fish continued to develop a full set of armor plates, with minimal changes in plate morphology (n = 8). (**C**) Implantation of Wnt-3a beads results in hypermorphic growth and armor plate fusion in the regions surrounding the exogenous Wnt-3a signal (n = 11). (**D**) Conversely, beads soaked in the Wnt inhibitor Dkk-1 inhibit plate formation surrounding the site of bead implantation (n = 10). Scale bar in **D** is 2 mm long.

implantation (*Figure 6C,D*). In contrast, implantation of Wnt-3a protein beads failed to produce a similar strong ring of GFP expression in transgenic fish carrying the mutated p3.2mar(T → G)-GFP construct (*Figure 6E,F*). Unexpectedly, the p3.2mar(T → G)-GFP construct did show a novel GFP response to the cyanoacrylate glue used in the implantation procedure, which was not seen in fish carrying p3.2mar-GFP. This expression was also observed in control manipulations with PBS beads (*Figure 6E*) or cyanoacrylate glue alone (data not shown), and was therefore distinct from the strong Wnt-3a response observed only with the fish carrying the p3.2mar-GFP construct.

Canonical Wnt signaling normally activates gene expression through changes in the activity of β-catenin (*Logan and Nusse, 2004*). Cotransfection of a β-catenin expression construct (pRK5-sk-βcatΔGSK) with the marine *EDA* enhancer driving a *luciferase* reporter (p3.2mar-*Luc*) produced a significant, dose-dependent increase in *luciferase* expression in cultured human keratinocytes in vitro (*Figure 6G*). Engineering the single SNP change in the marine enhancer (p3.2mar(T → G)-*Luc*) reduced but did not eliminate response to β-catenin in the heterologous system (28% lower expression with 50 ng of β-catenin, p < 0.001, n = 4).

Our combined experiments show that Wnt signaling can alter armor plate development and *EDA* expression in sticklebacks. The *EDA* enhancer region from high-plated sticklebacks also responds to Wnt signaling, while the single base pair mutation shared between NAKA and other low-plated sticklebacks significantly reduces Wnt responsiveness both in vivo and in vitro.

## Discussion

Previous work has shown that repeated armor plate reduction in sticklebacks is due in large part to genetic changes in the *EDA* region, though the causative molecular lesion(s) remained unknown (*Colosimo et al., 2005*; *Jones et al., 2012*). Our allele-specific expression experiments show that the freshwater allele of *EDA* is expressed at lower levels than the marine allele in F1 hybrids, confirming prior suggestions that there were likely to be *cis*-acting regulatory differences between marine and freshwater *EDA* variants (*Colosimo et al., 2005*). In addition, we have now identified a specific enhancer region in the key armor plates region, shown that the marine version of this enhancer normally drives expression in developing armor plates, and identified a specific T → G base pair change within the

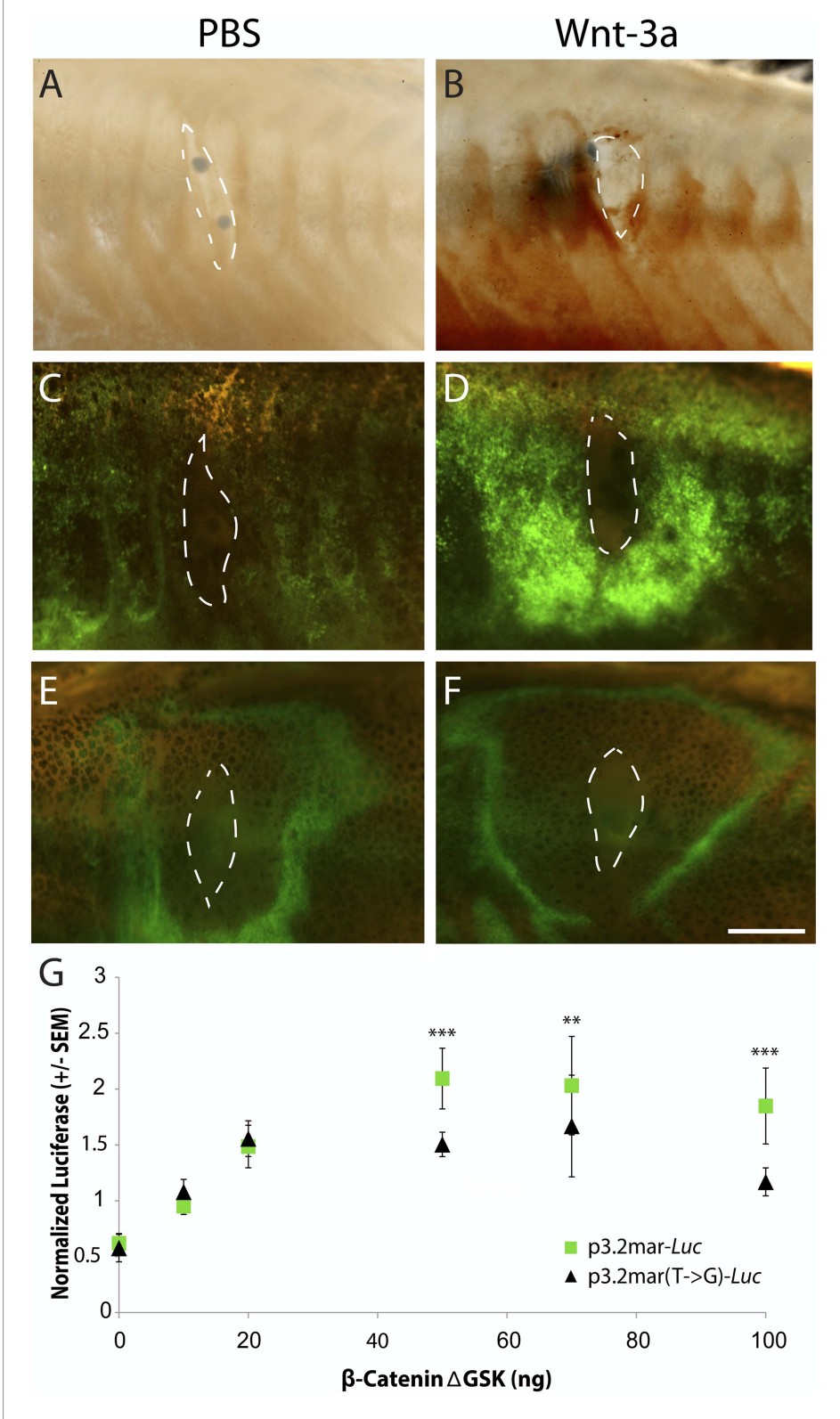

**Figure 6**. Single point mutation alters Wnt responsiveness of the *EDA* plate enhancer. Beads soaked in either PBS or Wnt-3a protein were implanted in the flanks of 2-month-old (24 mm long) marine fish. All images were taken at 48 hr post bead implantation. (**A**, **B**) RNAscope in situ hybridization for *EDA* expression induced by control bead placement (**A**) or Wnt-3a protein (**B**). The addition of Wnt-3a beads induces a ring of *EDA* expression (brown color in **B**)

*Figure 6. Continued*

directly surrounding the implantation site. (**C**, **D**) Bead implantation into the stable p3.2mar-GFP transgenic fish line. Control beads fail to induce GFP activity (**C**), whereas Wnt-3a beads induce a strong GFP response, seen in a ring surrounding the bead implantation site (**D**). (**E**, **F**) Bead implantation into the stable p3.2mar(T → G)-GFP line of transgenic fish. A ring of GFP expression is only seen at a distance from the implantation site of either control (**E**) or Wnt-3a (**F**) beads, corresponding to the location where cyanoacrylate glue was placed following implantation. Strong expression immediately surrounding the bead is not seen with Wnt-3a beads, in contrast to the result seen with p3.2mar-GFP transgenic fish (compare panels **F** and **D**). Scale bar in **F** is 1 mm long. (**G**) In vitro analysis of enhancer response to Wnt signaling via β-catenin co-transfection shows a strong induction of p3.2mar-*Luc* (green squares) with 50 ng or more of β-catenin in human HaCaT keratinocyte cells. The β-catenin-responsiveness of the p3.2mar(T → G)-*Luc* is significantly lower (black triangles). Combined p-values were calculated using Meta-P (***p < 0.001, **p < 0.01).

enhancer that is shared by all sequenced low-plated freshwater fish. Experimental recreation of the T → G base pair change reduces both armor plate expression and Wnt responsiveness of the enhancer, suggesting that this specific DNA change is the likely causative regulatory lesion in the *EDA* locus that leads to low-plated phenotypes in sticklebacks.

Like other genes found to underlie major morphological differences between marine and freshwater fish (*Shapiro et al., 2006*; *Miller et al., 2007*; *Chan et al., 2010*), the *EDA* gene is a key developmental control gene that is essential for formation of multiple tissues. Mutations in the coding region of *EDA* in both zebrafish and medaka cause deleterious phenotypes at multiple body sites, including complete loss of scales, partial loss of fins and teeth, and multiple craniofacial abnormalities (*Harris et al., 2008*; *Iida et al., 2014*). In contrast, the T → G regulatory change we have identified in an *EDA* enhancer leads to partial loss of *EDA* expression, particularly in the posterior flank region (*Figure 4*). This regulatory change thus alters *EDA* expression at the same body site where freshwater fish lack body armor, while preserving important functions of *EDA* in other tissues. These results provide a new example of a specific regulatory change linked to morphological evolution in natural populations (*Martin and Orgogozo, 2013*), and add to growing evidence that regulatory changes are a predominant mechanism underlying adaptive evolution in sticklebacks (*Jones et al., 2012*) and other organisms (*Wray, 2007*; *Carroll, 2008*).

Our results also provide new insight into genomic mechanisms contributing to repeated evolution. Previous analyses identified a shared low-plated *EDA* haplotype that has been fixed in most low-plated Pacific and Atlantic freshwater populations, and that is also present at very low frequency in the heterozygous state in marine populations (*Colosimo et al., 2005*; *Barrett et al., 2008*; *Bell et al., 2010*). Thus, *EDA* has become a classic example of rapid parallel evolution based on a preexisting genetic variant that increases in frequency when marine populations colonize new freshwater environments (*Stern, 2013*). The current results suggest that repeated evolution of low-plated phenotypes might also result from independent mutations occurring in the *EDA* locus in different populations. In previous surveys, Japanese NAKA fish were the only low-plated freshwater population that did not share the same *EDA* haplotype as other freshwater populations (*Colosimo et al., 2005*). Our experiments show that NAKA and other freshwater sticklebacks share an identical T → G non-coding regulatory mutation that reduces expression of *EDA* specifically in developing posterior armor plates. Characteristic flanking SNPs are not shared between NAKA and other low-plated populations, suggesting that the same T → G mutation has likely occurred independently on two very different haplotypes.

Recurrent mutations can be due to a particular DNA sequence that has a high intrinsic mutation rate. For example, previous studies of pelvic reduction in sticklebacks suggest that a key pelvic enhancer repeatedly deleted in freshwater populations has sequence features shared with fragile sites in human chromosomes (*Chan et al., 2010*). Individual base pairs can also be prone to particular mutations. For example, C → T transition mutations are particularly common at CpG dinucleotides in mammalian genomes, due to a high rate of spontaneous deamination of methylated C residues (*Mancini et al., 1997*; *Xia et al., 2012*). In contrast, the recurrent regulatory mutation we have identified at the stickleback *EDA* locus is a T → G transversion substitution, one of the least common types of changes seen in large scale studies of spontaneous germ-line mutations in humans

(*Kong et al., 2012*; *Genome of the Netherlands Consortium, 2014*) as well as in flies, worms, and yeast (*Lynch, 2010*).

It is possible that the shared T → G change arose not by independent mutation, but by extensive recombination or gene conversion from the typical freshwater *EDA* haplotype. Migratory marine populations include rare individuals that are heterozygous for both marine and freshwater haplotypes, which likely arise by repeated rounds of introgression of freshwater alleles into marine populations (*Colosimo et al., 2005*; *Schluter and Conte, 2009*). Sequence studies suggest that recombination can occur between typical marine and freshwater haplotypes, producing smaller and smaller blocks of sequence shared among most low-plated populations (*Colosimo et al., 2005*). In previous studies, the minimal shared freshwater region was approximately 16 kb, consisting of regions of both the *EDA* gene and two flanking genes involved in immune functions (*Colosimo et al., 2005*). However further recombination between marine and freshwater haplotypes could narrow this region further, conceivably approaching the size of a single base pair. For example, we have recently surveyed 263 migratory marine sticklebacks from Alaska and identified 12 completely plated individuals that are heterozygous for the T → G change in the *EDA* enhancer (minor allele frequency 2.3%). Analysis of flanking SNPs suggests one of these carriers is heterozygous for a larger characteristic freshwater haplotype, three are heterozygous for a much shorter freshwater haplotype, and eight are heterozygous at the T → G position but are marine-like at other characteristic flanking SNPs tested (*Supplementary file 1*). These data show that migratory marine populations can carry freshwater haplotypes of different sizes, including much smaller regions surrounding the key T → G regulatory change. Although most low-plated populations have clearly fixed a multi-kilobase haplotype surrounding *EDA*, the large size of this haplotype may reflect co-selection for additional phenotypes controlled by the closely linked genes (*Colosimo et al., 2005*). The geographically distant NAKA population is low-plated but shares only the T → G change, either because of an independent mutation, or because of fixation of a tiny fragment of the typical *EDA* haplotype. The NAKA population may be useful in the future for distinguishing the phenotypic effects of the isolated T → G regulatory change versus the larger *EDA* haplotype typically found in most low-plated sticklebacks.

The absence of a greater range of armor plate mutations at the *EDA* locus could be due to the relatively high frequency of a preexisting freshwater haplotype, whose frequency in migratory populations exceeds the rate of many spontaneous mutations. Alternatively, the T → G change could represent one of very few possible ways of producing a major change in armor plate patterns while still preserving other functions of the *EDA* gene. A constrained spectrum of mutations has been observed in other contexts involving very specific phenotypes. For example, nearly all patients with classic achondroplasia contain the same Gly380Arg (G → C) substitution in FGFR3 (*Horton et al., 2007*). This Gly380Arg substitution leads to a constitutively active FGF receptor that is thought to confer a selective advantage to spermatogonial cells (*Tiemann-Boege et al., 2002*; *Choi et al., 2008*). Identical amino acid substitutions in particular genes also underlie several examples of repeated evolution including insecticide resistance in insects (*GABA*), tetrodotoxin resistance in snakes (*NaK-ATPase*), C4 photosynthesis in plants (*PEPC*), and dark pigmentation in mice and birds (*MC1R*) (*Stern, 2013*). These and other well-studied cases typically involve particular amino acid changes that alter protein activity in specific ways, rather than completely ablating protein function. In contrast, few examples are known of identical recurrent base pair mutations in non-coding regulatory sequences (*Martin and Orgogozo, 2013*), though multiple cases are now being uncovered in large-scale sequencing surveys of replicate microbial evolution (*Tenaillon et al., 2012*; *Blank et al., 2014*). In a recent large-scale study of parallel temperature adaptation over 2000 generations, recurrent use of particular genes was at least 10 times more common than recurrent use of the same base pair changes within those genes (*Tenaillon et al., 2012*). Of the relatively rare recurrent base pair changes, those affecting protein-coding sequence also outnumbered those affecting non-coding intergenic sequence by nearly threefold. The T → G base pair change we have identified near *EDA* provides a rare example in vertebrates of a particular non-coding base pair change contributing to repeated adaptive evolution.

Our experiments also show that Wnt signaling acts upstream of *EDA* control sequences in armor plate patterning, and that the low-plated SNP reduces Wnt responsiveness of the *EDA* enhancer (*Figures 5, 6*). Although canonical Wnt signaling typically acts through the β-catenin and Lef transcription factors, the particular T → G base pair change we have identified does not alter a canonical Lef binding sequence. However, Wnt signaling is known to interact with multiple additional

signaling and transcription factor pathways, and the T → G change does alter a predicted binding site for c-Jun in the marine sequence (*Newburger and Bulyk, 2009*), which can act in collaboration with Wnt signaling in chondrocyte dedifferentiation (*Hwang et al., 2005*), osteopontin promoter activation in mammary cells (*El-Tanani et al., 2004*), and complexes with β-catenin to bind the promoters of Wnt target genes both in mammalian cells and in zebrafish (*Gan et al., 2008*). There are seven base pair positions in the predicted marine c-Jun binding site, and 21 corresponding single bp mutations that could alter one of these bases. 19 of these potential mutations are predicted to eliminate c-Jun binding (*Messeguer et al., 2002*). Of these 19 mutations, the T → G change found in low- plated sticklebacks is the only mutation that is also predicted to create a new overlapping binding site for AP-2α in the low-plated sequence. AP-2α has been shown to inhibit Wnt signaling by complexing with APC/β-catenin (*Li and Dashwood, 2004*; *Li et al., 2009b*). A new binding site for AP-2α could contribute to the reduced Wnt responsiveness of the freshwater *EDA* gene, or may contribute to other novel expression patterns that are not yet understood (such as the enhanced cyanoacrylate response we have observed with the T → G mutated enhancer). Future experiments are needed to test whether c-Jun, AP-2α or other factors interact directly with the *EDA* enhancer of either marine or freshwater sticklebacks. However, the simultaneous loss and gain of specific binding sites is a good example of the type of dual molecular constraints that could limit the range of possible base pair substitutions found underlying adaptive regulatory evolution at the *EDA* locus.

Our findings that connect Wnt signaling, plate development, and EDA signaling in sticklebacks also suggest new candidates for trans-acting genetic factors that may modify armor plate number in evolving populations. Previous genetic studies have shown that while the majority of the variance (>75%) in armor plate number in stickleback crosses maps to the *EDA* locus, the remainder of the variance can be explained by multiple plate modifier loci located on other chromosomes (*Colosimo et al., 2004*). Interestingly, two of the three previously mapped armor plate modifier regions contain genes for members of the Wnt pathway: *WNT11* on chromosome VII and β-catenin (*CTTNB1*) on chromosome X. Given the dramatic effects of Wnt signaling on armor plate development and *EDA* regulation (*Figures 5, 6*), these or other components of the Wnt signaling pathway are strong candidates for additional loci that may contribute to the adaptive fine-tuning of armor plate numbers that is known to occur in many low-plated populations (*Hagen and Gilbertson, 1972*, *1973*; *Moodie, 1972*; *Moodie et al., 1973*; *Bell and Haglund, 1978*).

## Materials and methods

### Allele-specific expression

Allele-specific expression differences were detected using pyrosequencing analysis of F1 hybrid fish as previously described (*Wang and Elbein, 2007*). In brief, a marine female from Rabbit Slough, AK was crossed to a freshwater benthic male fish from Paxton Lake, British Columbia to generate F1 hybrids that were heterozygous for a SNP in the *EDA* gene. Hybrid fish were raised to 13 mm standard length, a stage where the first few armor plates are forming in anterior tissues, but posterior plates have not yet formed. Multiple tissues were dissected, including: first dorsal spine, second dorsal spine, pelvic spines, pectoral fins, caudal fin, dorsal fin, anal fin, premaxilla with oral teeth, lower jaw (approximately the articular and dentary with oral teeth), left anterior flank skin between the second dorsal spine and pelvic spine (where anterior plates are forming), and left posterior flank skin between the dorsal fin and anal fin (where posterior plates will later form). RNA was prepared from dissected tissues using the TRI Reagent Protocol (Life Technologies, Carlsbad, CA). cDNA was synthesized using the Superscript III Supermix (Life Technologies) with random hexamer primers. A 183 bp product from the *EDA* gene was amplified using a biotinylated forward primer 5′-TCCACCAGAAGCGGGATACA-3′ and the reverse primer 5′-TTATGCCCCGGTTATCCTGTG-3′. Amplified products were sequenced using the primer 5′-TCTCCTCATGACCCTCT-3′, and the percentage of the two SNP alleles was calculated by EpigenDx, Inc. (Hopkinton, MA).

### DNA sequence comparisons

The 16 kb *EDA* candidate interval from NAKA fish was amplified as several long PCR products and assembled using Sanger sequencing (GenBank entry KP164994). Alignment of the NAKA sequence with the complete sequence of the *EDA* region from Salmon River (SALR) marine and Paxton Benthic (PAXB) freshwater BAC clones (*Colosimo et al., 2005*); and the Bear Paw Lake (BEPA) reference

genome (*Jones et al., 2012*); identified 13 positions where low-plated NAKA, PAXB, and BEPA differed from high-plated SALR fish. Reexamination of these positions in sequence reads from 21 marine and freshwater genomes (*Jones et al., 2012*) placed with SAMtools (*Li et al., 2009a*) against the BEPA reference genome, and resequencing of additional fish, identified the chrIV:12811481 position as shared among all low-plated sticklebacks examined. Population codes and source locations are as previously described (*Colosimo et al., 2005*; *Jones et al., 2012*).

The region surrounding the T → G base pair change was subsequently amplified from 263 fully plated migratory sticklebacks collected from Rabbit Slough, AK (RABS), using 5′-TTGACAAGTGATGTTCTCTGTGGCC-3′ and 5′-ATGTTGGAGCTGGCAGGAGGAGG-3′. All heterozygous carrier fish were then tested for the characteristic flanking SNPs previously found to distinguish most high-plated and low-plated haplotypes in previous studies (*Colosimo et al., 2005*). SNPs 5 and 6 at positions 12808303 and 12808630 were determined by amplifying and sequencing a genomic region using 5′-CAGAGGAGGTGAAACCGCACTTACA-3′ and 5′-TGGGAACGCGTCGACATTTGGGA-3′. SNP 7 at position 12811933 was called from the same genomic amplification used to recover the T → G regulatory change. SNPs 8 and 9 at positions 12813328 and 12813394 were determined by amplifying and sequencing a genomic region using 5′-GTGCCCAGGAGCTCTAGACTTGGC-3′ and 5′-TCTCACATCCGGCAGCGACAAGC-3′.

## Plasmids

The plate enhancer region was amplified from genomic DNA of a marine fish from Salmon River, British Columbia using 5′-ATGTGGCCAGATAGGCCACATGGTGTGGGAGAGCAGTGATCG-3′ and 5′-ATGTGGCCTATCTGGCCATGTTGGAGCTGGCAGGAGGAGG-3′ primers that each contain SfiI linkers. The 3.2 kb amplified fragment was cloned into the SfiI site of the pT2HE GFP reporter vector (modified from *Kawakami, 2007*) to generate p3.2mar-GFP.

Site directed mutagenesis was performed on the p3.2mar-GFP plasmid to induce a single freshwater base pair change using two 40 bp overlapping primers 5′-AATTAGTTCCATCTTGAGAGGCAGAGAGAAGATGGTTCCT-3′ and 5′-AGGAACCATCTTCTCTCTGCCTCTCAAGATGGAACTAATT-3′. A 15-cycle PCR amplification using 50 ng of plasmid, 125 ng of primers, and Phusion polymerase was performed to induce the base pair change (*Zheng et al., 2004*). The resulting plasmid, p3.2mar (T → G)-GFP, was verified by DNA sequencing.

For cell culture experiments, the enhancer inserts from p3.2mar-GFP and p3.2mar(T → G)-GFP were excised from the pT2He plasmid using SfiI and cloned into the XhoI site of the pTA-*Luc* vector (Clontech Laboratories, Inc., Mountain View, CA), to generate p3.2mar-*Luc* and p3.2mar(T → G)-*Luc*. The β-catenin expression plasmid pRK5-sk-βcatΔGSK was a gift from the Nusse Lab.

## Transgenic enhancer assays

Transgenic sticklebacks were generated by microinjection of freshly fertilized eggs as previously described (*Chan et al., 2010*). Plasmids were co-injected with *Tol2* transposase mRNA as described (*Fisher et al., 2006*; *Wada et al., 2010*). Mature *Tol2* mRNA was synthesized by in vitro transcription using the mMessage mMachine SP6 kit (Life Technologies). All enhancer assays were performed on high-plated fish derived from Little Campbell River (British Columbia), Bodega Bay (California), or Rabbit Slough (Alaska). Microscopic observation for GFP expression was conducted with a MZFLIII fluorescent microscope (Leica Microsystems, Bannockburn, IL) using GFP2 filters and a ProgResCF camera (Jenoptik AG, Jena, Germany) to distinguish GFP expression from autofluorescence in pigmented fish. We generated stable lines by making crosses from mosaic founder animals.

## Whole-mount RNAscope in situ hybridization

Two-month-old fish (20–24 mm standard length) were fixed in 4% paraformaldehyde overnight at 4℃, washed, and stored in methanol at −20℃ for up to 6 months prior to in situ hybridization. Fish were rehydrated through a series of methanol/water washes (90%, 75%, 50%, 25%, 0), bleached in 6% hydrogen peroxide rocking at room temperature for up to 3 hr, and treated with 10 μg/ml Proteinase K in water rocking for 7.5 min in order to detect *EDA* signal. From this point, the RNAscope Brown Protocol was followed with an *EDA* probe designed by Advanced Cell Diagnostics (Hayward, CA) with two procedural modifications: Pretreatment 2 was performed at 40℃ and the hybridization step with *EDA* probe was allowed to proceed overnight (*Wang et al., 2012*; *Gross-Thebing et al., 2014*).

## Bead experiments

Affi-Gel Blue Gel beads (BioRad Laboratories, Inc., Hercules, CA) were soaked overnight in PBS, 1.2 μg of recombinant human Wnt-3a (R&D Systems, Minneapolis, MN), or 1.2 μg of recombinant mouse Dkk-1 (R&D Systems). Marine-derived fish were raised to 20–24 mm standard length (first four armor plates present), anesthetized with Tricaine (0.017% wt/vol), and an average of 12 beads were placed into the flank of each fish, posterior to the apparent plates. Cyanoacrylate glue (Loctite Super Glue) was used to close the skin surrounding the implantation site. Fish were allowed to recover for 48 hr before further experimentation or to continue developing into adulthood. The beads' effects on overall plate development were analyzed in live adult fish using 0.2% Calcein in aquarium water to mark newly ossified bones as previously described (*Kimmel et al., 2003*; *Wada et al., 2010*).

## Cell culture experiments

HaCaT human keratinocyte cells (*Boukamp et al., 1988*) were cultured in DMEM supplemented with 10% FBS, 2 mM L-glutamine and 1% penicillin-streptomycin. Cells were seeded into 24-well plates at a density of $1 \times 10^5$ cells/well and transfected after 24 hr. 300 ng of p3.2mar-*Luc* or p3.2mar(T → G)-*Luc* plasmids were cotransfected together with 0–100 ng of pRK5-sk-βcatΔGSK using Lipofectamine 2000 (Life Technologies) according to manufacturer's protocol. After 6 hr of transfection, cell culture medium was replaced with standard medium supplemented with 2.8 mM calcium chloride (Sigma, St. Louis, MO). Cell lysates were collected after 48 hr and assayed using the Dual-Luciferase Reporter Assay System (Promega, Madison, WI) according to the manufacturer's instructions.

## Acknowledgements

We thank Seiichi Mori and Michael Bell for samples of NAKA and RABS fish; Tim Howes for the pT2HE plasmid; Roel Nusse's lab for the pRK5-sk-βcatΔGSK plasmid; Mike McLaughlin for fish care; and Roel Nusse, Will Talbot, Sarah McMenamin, Zach O'Brown and members of the Kingsley lab for useful discussions. This research was supported in part by a National Institutes of Health Training Grant (NIGMS T32 GM007790, NO'B), National Science Foundation Graduate Research Fellowships (NO'B, BS), and a NIH Center of Excellence in Genomic Science grant (NHGRI 3P50 HG002568, DMK). DMK is an investigator of the Howard Hughes Medical Institute.

## Additional information

### Funding

| Funder | Grant reference number | Author |
| --- | --- | --- |
| National Science Foundation (NSF) | | Natasha M O'Brown, Brian R Summers |
| Howard Hughes Medical Institute (HHMI) | | Shannon D Brady, David M Kingsley |
| National Institute of General Medical Sciences (NIGMS) | T32 GM007790 | Natasha M O'Brown |
| National Human Genome Research Institute (NHGRI) | 3P50 HG002568 | Felicity C Jones, David M Kingsley |

The funders had no role in study design, data collection and interpretation, or the decision to submit the work for publication.

### Author contributions

NMO'B, Conception and design, Acquisition of data, Analysis and interpretation of data, Drafting or revising the article; BRS, Conception and design, Acquisition of data, Analysis and interpretation of data; FCJ, DMK, Conception and design, Analysis and interpretation of data, Drafting or revising the article; SDB, Acquisition of data, Drafting or revising the article

### Ethics

Animal experimentation: This study was performed in accordance with the recommendations in the Guide for the Care and Use of Laboratory Animals of the National Institutes of Health. All of the

animals were handled according to approved institutional animal care and use committee (IACUC) protocols (#13834) of Stanford University, in animal facilities accredited by the Association for Assessment and Accreditation of Laboratory Animal Care International (AAALAC).

## Additional files

### Supplementary file

• Supplementary file 1. *EDA* haplotypes found in typical marine and freshwater populations, and in rare heterozygous carrier fish from Alaska. Previous studies have identified characteristic base pair positions that differentiate the most abundant *EDA* haplotypes present in marine and freshwater populations. The position of these SNPs in the *EDA* region are shown, along with the exon number in which they occur, the nature of the mutation (S: synonymous change; N: non-synonymous change; Reg: regulatory change), the SNP number assigned in previous studies, and the characteristic alleles that are found in 10 different completely-plated marine (red) and 14 different low-plated freshwater (blue) populations (top two haplotype rows, see *Colosimo et al., 2005*). We amplified the genomic region surrounding the new T → G regulatory mutation identified in the current study, and found that 12 of 263 completely plated migratory fish from Rabbit Slough, Alaska (RABS) were heterozygous for the T → G change (minor allele frequency: 2.3%). Analysis of flanking positions suggests that these carrier fish are heterozygous for larger or smaller haplotypes at the *EDA* locus, or are marine-like at all positions except for the T → G change like NAKA. The six indicated SNPs correspond to positions chrIV: 12808303, 12808630, 12811481, 12811933, 12813328, 12813394 in the stickleback genome assembly (*Jones et al., 2012*).

### Major dataset

The following previously published dataset was used:

| Author(s) | Year | Dataset title | Dataset ID and/or URL | Database, license, and accessibility information |
|---|---|---|---|---|
| Jones et al., | 2012 | Data from: The genomic basis of adaptive evolution in threespine sticklebacks (*Nature* **484**: 55-61) | http://sticklebrowser.stanford.edu | Publicly available. |

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
