## [Decision Letter]

Thank you for sending your work entitled “A recurrent regulatory change
underlying altered expression and Wnt response of the stickleback armor plates gene
*EDA*” for consideration at *eLife*. Your
article has been favorably evaluated by Detlef Weigel (Senior editor), Robb Krumlauf
(Reviewing editor) and three reviewers.

The Reviewing editor and the reviewers discussed their comments before we reached this
decision, and the Reviewing editor has assembled the following comments to help you
prepare a revised submission.

The work is significant because it lends new biological insight into the molecular
mechanism of adaptive evolution in natural populations and also expands our knowledge of
the regulatory landscape that controls *EDA* expression. This finding has
both developmental and potential clinical significance. More generally, this work is
remarkable in that it is also a rare instance in which a non-coding change appears to
have repeatedly been the target of selection in multiple populations. It would have
enhanced the study to know which factors bind to this region, or data on the phenotypic
consequences of engineering a correction of this change, or of mutating the same base,
but to a different substitution. Since such additional work would require a good deal
more time and effort, on balance the reviewers decided it would be beyond the scope of
this study to require such additional data.

However, we would like to ask you to provide more information on the status of the T to
G change in a larger number of marine animals and to tone down your inference that there
might be very few, perhaps only one mutation that is evolutionarily permissible, unless
you can demonstrate with additional mutations in the enhancer that this is indeed the
case.

Also, in your previous work, you did not distinguish between gene-flow from freshwater
into migratory marine fish and ancestral presence in the marine population, nor did you
explore potential reasons for the remarkable size of the common low-plated 16 kb
haplotype. Please add relevant information and address these questions.

---

## [Author Response]

*The work is significant because it lends new biological insight into the
molecular mechanism of adaptive evolution in natural populations and also expands our
knowledge of the regulatory landscape that controls* EDA *expression.
This finding has both developmental and potential clinical significance. More
generally, this work is remarkable in that it is also a rare instance in which a
non-coding change appears to have repeatedly been the target of selection in multiple
populations. It would have enhanced the study to know which factors bind to this
region, or data on the phenotypic consequences of engineering a correction of this
change, or of mutating the same base, but to a different substitution. Since such
additional work would require a good deal more time and effort, on balance the
reviewers decided it would be beyond the scope of this study to require such
additional data*.

*However, we would like to ask you to provide more information on the status of
the T to G change in a larger number of marine animals*.

We have now carried out a PCR and sequencing screen to look for the presence of the T to
G mutation in a population of migratory marine fish from Rabbit Slough AK. (This work
was carried out in part by Shannon Brady, who has now been added a co-author of the
manuscript). This survey showed that most completely plated marine animals are
homozygous for the marine “T” allele, but a small number of fish are
heterozygous carriers of the freshwater “G” allele (overall minor G allele
frequency: 2.3%). We also typed these heterozygous fish with flanking markers, to
distinguish between carriers of large, small, or recombinant freshwater
*EDA* haplotypes. These data are summarized in a new [Supplementary-material SD1-data], and a
corresponding section has been added to both the Discussion (paragraph 5) and the
Materials and methods (paragraph 3) to cover the new material.

*Tone down your inference that there might be very few, perhaps only one mutation
that is evolutionarily permissible, unless you can demonstrate with additional
mutations in the enhancer that this is indeed the case*.

We have examined the location of the marine and freshwater sequence change for predicted
transcription factor binding sites, and computationally compared the predicted effects
of all possible single base pair mutations at the positions surrounding the observed
T->G change. While there are multiple mutations that can potentially disrupt the
c-Jun site previously mentioned in the Discussion, the observed T->G change is
the only base pair mutation that is predicted to both eliminate a c-Jun site in the
marine enhancer, and to simultaneously create a new overlapping AP-2 alpha site in the
freshwater sequence. We describe these predicted mutational effects in an enlarged
paragraph 7 of the Discussion, and discuss how the dual effects of the T->G
mutation could be an example of combined molecular constraints that limit the range of
base pair mutations observed in low-plated populations.

We have also edited the Discussion to clearly state the possibility that the observed
sharing of T->G changes is due to the relatively high frequency of the freshwater
G allele in marine populations, rather than due to strong constraints on possible
adaptive sequence changes among independently arising mutations (paragraph 5 and first
sentence of paragraph 6).

*Also, in your previous work, you did not distinguish between gene-flow from
freshwater into migratory marine fish and ancestral presence in the marine
population*.

This distinction was not discussed in the original Colosimo et al. paper in 2005.
However, the issue has subsequently been further analyzed using the geographic patterns
found in the Colosimo et al. marine and freshwater *EDA* allele
sequences. This analysis shows much greater geographic structuring among freshwater
*EDA* alleles than among marine *EDA* alleles, which is
consistent with repeated rounds of gene flow from freshwater fish into migratory marine
populations, rather than selection from ancestral diversity already present in marine
populations (55). We now
briefly refer to this point in the revised Discussion (see paragraph 5).

*Nor did you explore potential reasons for the remarkable size of the common
low-plated 16 kb haplotype*.

Our new survey of migratory marine fish includes additional analysis of freshwater
haplotype sizes in carrier animals, and identifies multiple examples of apparent
recombinant haplotypes that are smaller than the previous observed 16 kb freshwater
region ([Supplementary-material SD1-data]). We have also modified the Discussion to point out that the substantial
size of the typical 16 kb freshwater haplotype may be due to coselection for multiple
phenotypes arising from linked genes, and that NAKA may be an interesting example of a
population that has fixed only the armor plate changes (see the end of paragraph 5 of
the revised Discussion).